# Hippocampal seed connectome-based modeling predicts the feeling of stress

Elizabeth V. Goldfarb [1,2,3], Monica D. Rosenberg [4,5], Dongju Seo[1,2], R. Todd Constable [3,6] & Rajita Sinha [1,2,7✉]

Although the feeling of stress is ubiquitous, the neural mechanisms underlying this affective experience remain unclear. Here, we investigate functional hippocampal connectivity throughout the brain during an acute stressor and use machine learning to demonstrate that these networks can specifically predict the subjective feeling of stress. During a stressor, hippocampal connectivity with a network including the hypothalamus (known to regulate physiological stress) predicts feeling more stressed, whereas connectivity with regions such as dorsolateral prefrontal cortex (associated with emotion regulation) predicts less stress. These networks do not predict a subjective state unrelated to stress, and a nonhippocampal network does not predict subjective stress. Hippocampal networks are consistent, specific to the construct of subjective stress, and broadly informative across measures of subjective stress. This approach provides opportunities for relating hypothesis-driven functional connectivity networks to clinically meaningful subjective states. Together, these results identify hippocampal networks that modulate the feeling of stress.

[1] Yale Stress Center, Yale University School of Medicine, New Haven, CT 06519, USA. [2] Department of Psychiatry, Yale School of Medicine, New Haven, CT 06511, USA. [3] Department of Diagnostic Radiology and Biomedical Imaging, Yale School of Medicine, New Haven, CT 06520, USA. [4] Department of Psychology, Yale University, New Haven, CT 06520, USA. [5] Department of Psychology, The University of Chicago, Chicago, IL 60637, USA. [6] Department of Neurosurgery, Yale School of Medicine, New Haven, CT 06520, USA. [7] Department of Neuroscience, Yale School of Medicine, New Haven, CT 06520, USA. ✉email: rajita.sinha@yale.edu

F eeling "stressed out" is a common human experience. Although it is intuitively understood, the neurobiological mechanisms underlying this subjective experience remain unclear. High subjective stress is associated with negative long-term consequences for mental[1,2] and physical health[3–5], and immediate detrimental effects on cognition[6]. Yet expressing stress-related feelings can also be adaptive[7], and increasing awareness of feelings is a goal of emotion regulation and mindfulness-based stress reduction[8]. Thus, there is a need to understand how feelings of stress arise and the brain networks that underlie this core human experience.

Although research across species has elucidated neurobiological mechanisms supporting physiological stress responses, including activation of the hypothalamic-pituitary-adrenal axis leading to glucocorticoid release[9,10], subjective feelings of stress often diverge from glucocorticoid responses[11–13]. Therefore, we cannot assume that the same processes governing physiological stress explain the feeling of stress. However, converging evidence suggests that the hippocampus, known to reduce glucocorticoid release by inhibiting the hypothalamus[14], may also contribute to the feeling of stress. The hippocampus is sensitive to stressor exposure across species[15–17]. In rodents, the hippocampus is necessary for anxiety behavior[18–20]. Recent findings show that hippocampal neurons encoding memory for prior stressors were reactivated in stress-susceptible mice, and optogenetically activating these neurons increased avoidance behavior[21]. In humans, hippocampal volume is associated with life stress, emotion dysregulation[22], cardiovascular stress reactivity[23], and vulnerability for perceived stress[24]. Cognitively, the hippocampal system may contribute to the subjective feeling of stress by supporting memory retrieval, which can either augment[25] or diminish[26] acute stress responses. Impairments in hippocampal function could change stress reactions through generalizing from prior stressful contexts[10] and increasing reliance on habitual coping strategies[27]. However, the contributions of functional hippocampal connectivity to subjective stress remain unknown.

Here, we use functional magnetic resonance imaging (fMRI) to investigate whether changes in hippocampal connectivity resulting from a brief, sustained stressor could predict the feeling of stress. Using a within-subjects design, participants were exposed to a validated fMRI-based sustained exposure paradigm[28–30] involving blocks with an uncontrollable barrage of highly aversive and threatening images (Stressor) or neutral/relaxing images (Neutral), from which we computed stressor-induced changes in background hippocampal connectivity (Fig. 1a, b). Feelings of stress were assessed repeatedly using distinct affective dimensions of rating stress and arousal[31] (Fig. 1c). In addition to acute feelings, we measured chronic, non-stressor specific stress appraisals using the Perceived Stress Scale[1].

We tested how stressor-modulated hippocampal networks predict sustained emergent feelings of stress by developing a predictive modeling approach. Brief emotional responses can be predicted from patterns of univariate and multivariate fMRI signal[32–34] and connectome-based predictive modeling (CPM) provides a powerful tool to predict behavior from functional connectivity[35,36]. Thus, we developed seed connectome-based predictive modeling (sCPM) to link stressor-modulated hippocampal connectivity to the complex, sustained state of subjective stress. Analyses reveal that distinct patterns of hippocampal connectivity during a stressor predict enhanced and diminished feelings of stress.

## Results
### Stressor changes feelings of stress and hippocampal connectivity.
The fMRI-based stressor successfully evoked feelings of stress in our sample ($N = 60$, demographics in Supplementary Table 1). Sustained ratings of both arousal and stress were greater during the stressor relative to neutral, non-stressful image exposure (Fig. 1d; controlling for sex and condition order, A: $F_{1,58} = 120.57$, $p < 0.001$, partial $\eta^2 = .96$; S: $F_{1,58} = 166.51$, $p < 0.001$, partial $\eta^2 = 0.96$), and stress ratings increased significantly over time ($F_{2,116} = 3.23$, $p = 0.043$, partial $\eta^2 = 0.045$). Chronic subjective stress (PSS) did not correlate with these acute stress or arousal ratings (all $p > 0.25$).

To capture spontaneous, intrinsic fluctuations in functional connectivity, we used a background connectivity approach[37], which has identified hippocampal coupling dynamics contributing to long-term memory[38]. After limiting synchronized stimulus-evoked responses through regression and bandpass filtering[39], as well as regressing out other potential confounds ("Methods"), we correlated the timeseries from an anatomical hippocampal region of interest (seed ROI) with the timeseries from all voxels throughout the brain, separately for stressor and neutral conditions. As function and stress responsivity vary along the anterior/posterior gradient[40,41], we repeated this analysis using percentile-defined anterior/posterior hippocampus as seed ROIs (hereafter aHPC and pHPC; Fig. 1f). Comparing the resultant seed-based functional connectivity maps revealed a diffuse network including stressor-induced increases in connectivity with pre/postcentral gyrus, putamen, and dorsolateral prefrontal cortex (dlPFC) as well as decreased connectivity with amygdala, hypothalamus, and parahippocampal cortex (PHC, Fig. 1e, f; overlapping clusters shown in Supplementary Table 2).

We tested whether these hippocampal networks were simply responsive to stressors or were also able to predict the subjective feeling of stress. As feelings changed over time, we investigated whether mean stressor-induced hippocampal connectivity predicted mean stressor-induced ratings of stress and arousal as well as whether early hippocampal connectivity predicted later ratings. We also explored whether stressor-modulated networks could predict individual differences in chronic subjective stress (PSS).

### Hippocampal connectivity during stressor predicts feelings of stress.
We applied sCPM to understand which stressor-modulated hippocampal functional connectivity patterns predicted feelings of stress (Fig. 2a). Similar to CPM[42], this approach uses leave-one-out cross-validation (LOO-CV) to extract clusters from the hippocampal seed connectivity map (Fig. 1e, f) to build models predicting behavior. For each set of $N−1$ participants, Fisher-transformed hippocampal connectivity with each significant cluster (relative to baseline) was computed per participant and correlated (Spearman $r_s$) with feelings of stress (again, relative to baseline). Clusters that had above-threshold correlations with feelings of stress were separated into a positive network (predicting more stress) and a negative network (predicting less stress). Next, linear models were built relating mean hippocampal connectivity with each network to feelings of stress. These models were used to predict the left-out individual's feelings of stress, based on that individual's connectivity between the hippocampus and positive/negative networks during the stressor. Predictive model power was assessed by comparing model-predicted with true feelings of stress ($r_s$). Statistical significance was determined nonparametrically by comparing $r_s$ against the distribution of null $r_s$, derived from repeating the analysis 1000x with randomly-shuffled ratings[35]. All models had medium to large effect sizes when these analyses were repeated using tenfold cross-validation (although, as expected with tenfold vs. LOO-CV, the correlation coefficients were smaller; Supplementary Table 3). Seed-based CPM analyses revealed networks that predicted higher and lower

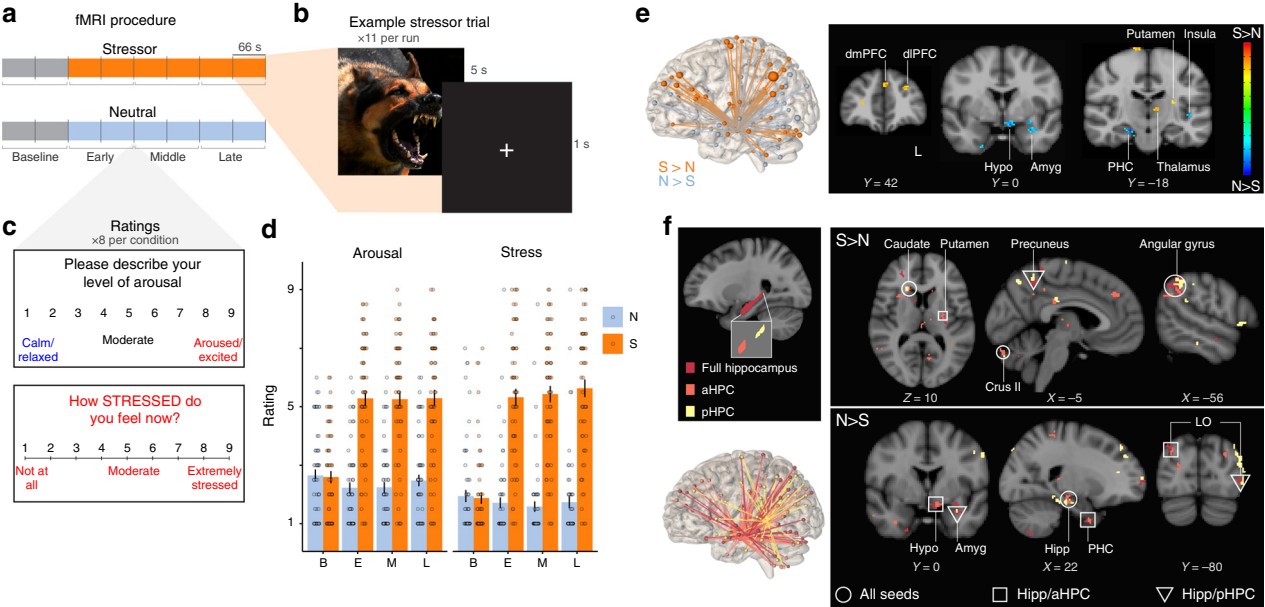

**Fig. 1 Stressor exposure influences feelings of stress and hippocampal connectivity. a** Design. Functional neuroimaging data (fMRI) were acquired as participants were exposed to 6 consecutive runs of neutral/relaxing (Neutral condition, N) or highly aversive images (Stressor, S) immediately preceded by fixation (Baseline, two runs). For analysis, these runs were aggregated into three two-run epochs (132 s each: Baseline, Early, Mid, Late), with mean connectivity and mean ratings computed within each of these windows. **b** In each condition, images (5 s) alternated with interstimulus intervals (1 s) during six 66-s runs. Photo shown is similar to those used in the experiment. **c** Ratings of feelings of stress were collected after each run on nine-point visual analog scales. Arousal levels were visualized using the Self-Assessment Manikin (not shown)[40]. **d** The stressor condition led to significant sustained increases in ratings of arousal and stress. B = baseline, E = early, M = Middle, L = late (as in **a**). Overlaid dots represent individual participants. Bars show mean values per epoch and condition across participants; error bars = $\pm$ 1SE. N = 60 participants. Source data are provided as a Source Data file. **e** Whole-brain hippocampal connectivity network significantly modulated by stressor exposure. Left, network overview, with node size reflecting number of voxels per identified cluster. Right, example slices from network. **f** Stressor-modulated connectivity networks from full, anterior, and posterior hippocampal seeds.

feelings of stress in novel individuals (Fig. 2b–g; Supplementary Table 4). Functional connectivity between the full hippocampus and hypothalamus and inferior temporal gyrus (ITG) throughout stressor exposure predicted higher overall arousal ratings. Even more remarkably, connectivity in early runs (first 2 min of stressor relative to baseline) predicted both higher (positive network) and lower (negative network) subsequent stress ratings (last 2 min). The positive network included hippocampal connectivity with hypothalamus, PHC, and medial temporal gyrus, whereas the negative network included connectivity with cerebellar vermis and postcentral gyrus/precuneus. In addition, negative networks predicting lower subsequent arousal ratings included early hippocampal connectivity with dlPFC, cerebellar vermis, and posterior putamen, and early pHPC connectivity with middle frontal gyrus, precentral gyrus, and cerebellum.

**Hippocampal networks adaptively respond during stressor.** The stressor-modulated hippocampal networks were defined as showing significantly different functional connectivity between the stressor and neutral conditions. This difference could have come from having either higher or lower connectivity during the stressor. We examined how the networks that predicted subjective stress were modulated by stressor exposure. This would indicate whether the stressor, on average, changed the hippocampal network consistent with amplifying feelings of stress (e.g., increasing connectivity in the network where higher connectivity predicted feeling more stressed), attenuating feelings of stress (e.g., decreasing connectivity in that same network), or had a random pattern (schematic in Fig. 3).

We first examined positive networks, in which higher connectivity predicted feeling more stressed. We found that the stressor decreased connectivity with each component of these

networks—consistent with attenuating feelings of stress. Strikingly, we observed the same adaptive pattern for negative networks, in which higher connectivity predicted feeling less stressed. For these networks, the stressor increased connectivity—which would also attenuate feelings of stress (Fig. 3, Z scores from group map in Supplementary Table 4). Post-hoc analyses suggest that participants with stress-related psychiatric histories ($N = 16$; Supplementary Table 1) showed less adaptive responses, particularly for hippocampus/hypothalamus connectivity, although this requires further evaluation in patient samples (Supplementary Fig. 1).

The above analyses reveal that stressor exposure modulated hippocampal networks in a pattern consistent with attenuated feelings of stress. Importantly, several of these networks were prospective—that is, changes in connectivity that occurred within the first 2 min of stressor exposure predicted feelings of stress measured at the end of stressor exposure. Although this temporal order suggests that hippocampal networks gave rise to these feelings, it is possible that these networks were more strongly linked to feelings measured concurrently (suggesting that feelings of stress might have preceded these connectivity changes). As an exploratory analysis, we used partial correlations[43] to test whether network strength was associated with subsequent feelings of stress, even when accounting for concurrent feelings of stress. Of the four networks that predicted subsequent stress, three continued to show significant associations with subsequent stress even with this control (hipp: Late stress [+]: $r_s = 0.42$, $p < 0.001$; Late arousal [−]: $r_s = -0.51$, $p < 0.001$; pHPC: Late arousal [−]: $r_s = -0.28$, $p = 0.03$). These analyses support the idea that these hippocampal networks responded during the stressor to influence subsequent subjective stress. Together, these results suggest an adaptive, multifaceted hippocampal connectivity

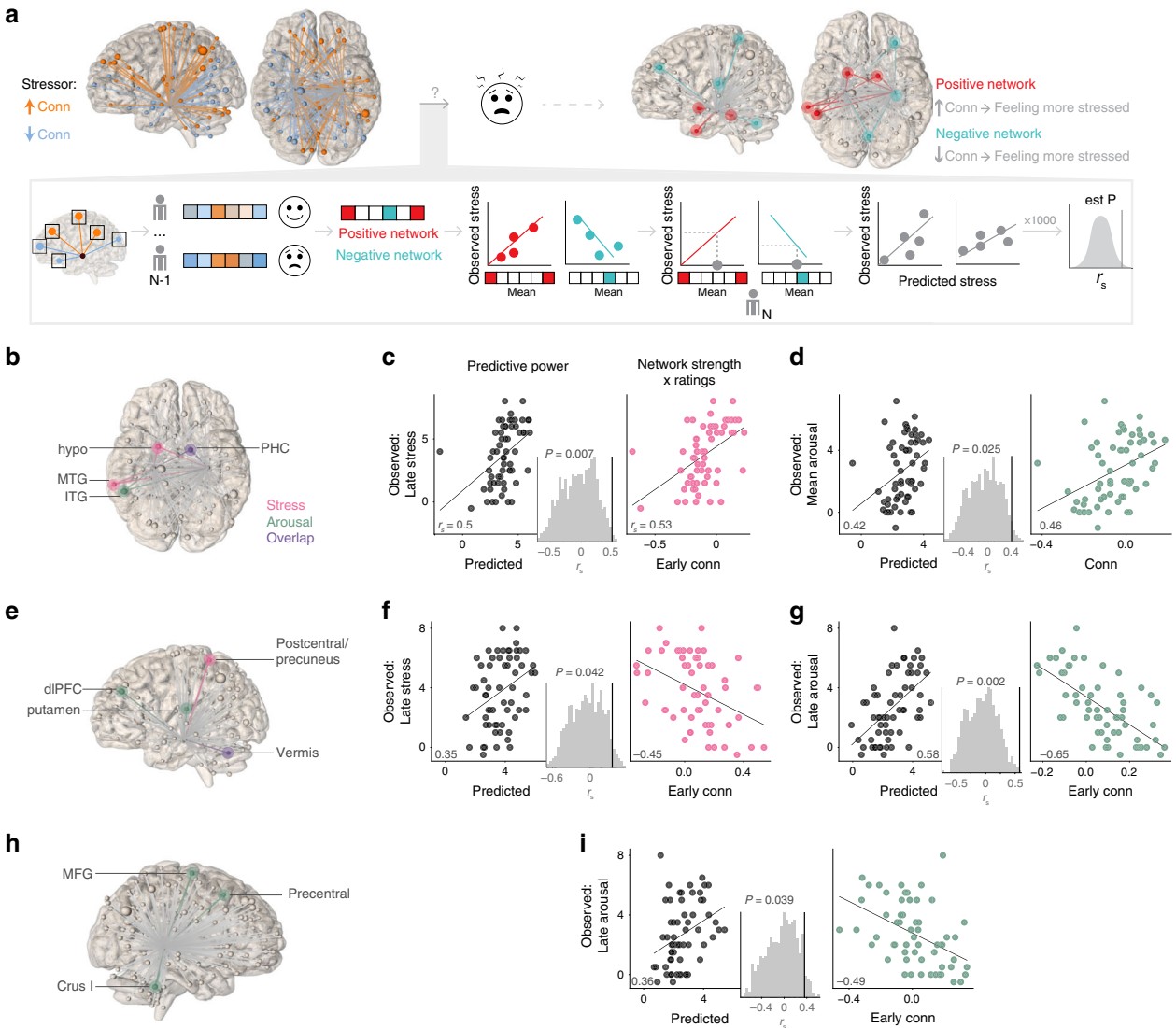

**Fig. 2 Stressor-induced hippocampal connectivity networks predict feelings of stress. a** Schematic of seed connectome-based predictive model (sCPM) analysis and summary of predictive networks. The goal of this analysis was to determine whether stressor-modulated hippocampal connectivity could predict feelings of stress and, if so, to identify the clusters that were part of positive (predict feeling more stressed) and negative (predict feeling less stressed) networks. We used leave-one-out cross-validation to determine which clusters correlated (positively and negatively) with feelings of stress across *N-1* participants, and used these linear models to predict feelings of stress in left-out participant *N*. **b–g** Hippocampal functional connectivity networks predict feelings of stress. **b–d** Positive networks separated by whether they predicted ratings of stress (pink) or arousal (green). **b** Anatomical distribution of positive network. Clusters that predicted both stress and arousal shown in purple. **c, d** Summary of network performance. Left, predictive power; Spearman's correlation ($r_s$) of model-predicted with observed ratings. Histogram shows distribution of correlation values from 1000 iterations of randomly-shuffled feelings of stress used to nonparametrically determine *P* values. Right, correlation between overlap network strength (i.e., mean of clusters selected on every leave-one-out iteration) and observed ratings. **e–g** Negative networks. As in (**b**), clusters that predicted both stress and arousal are shown in purple. **h, i** Posterior hippocampal functional connectivity network predicting lower arousal ratings. Source data for network performance are provided as a Source Data file.

response (extending across both positive and negative networks) that is engaged during a stressor to modulate the feeling of stress.

**Predictive hippocampal networks are specific.** Having shown that stressor-modulated hippocampal networks significantly predict feelings of stress, we next tested the specificity of these predictions. In particular, we investigated whether stressor-modulated hippocampal networks could selectively predict the construct of stress and whether the hippocampal network was especially informative for generating these predictions.

To examine construct specificity, we used subjective ratings of a construct unrelated to stress (focus) that was measured

concurrently with stress and arousal ratings (Fig. 4a). Overall, participants reported being highly focused on the images, and this did not significantly differ between conditions (Stressor vs. Neutral: $t_{59} = 1.54$, $p = 0.13$). We found that, even when hippocampal connectivity predicted feelings of stress, it did not predict ratings of focus (all models in Supplementary Fig. 2). An example model is shown in Fig. 4a (difference in correlation with stress vs. focus $Z = 1.87$, $p = 0.06$). These results demonstrate that predictive hippocampal networks are specific to the feeling of stress.

To test whether hippocampal connectivity was specifically informative about subjective stress, we tested the null hypothesis

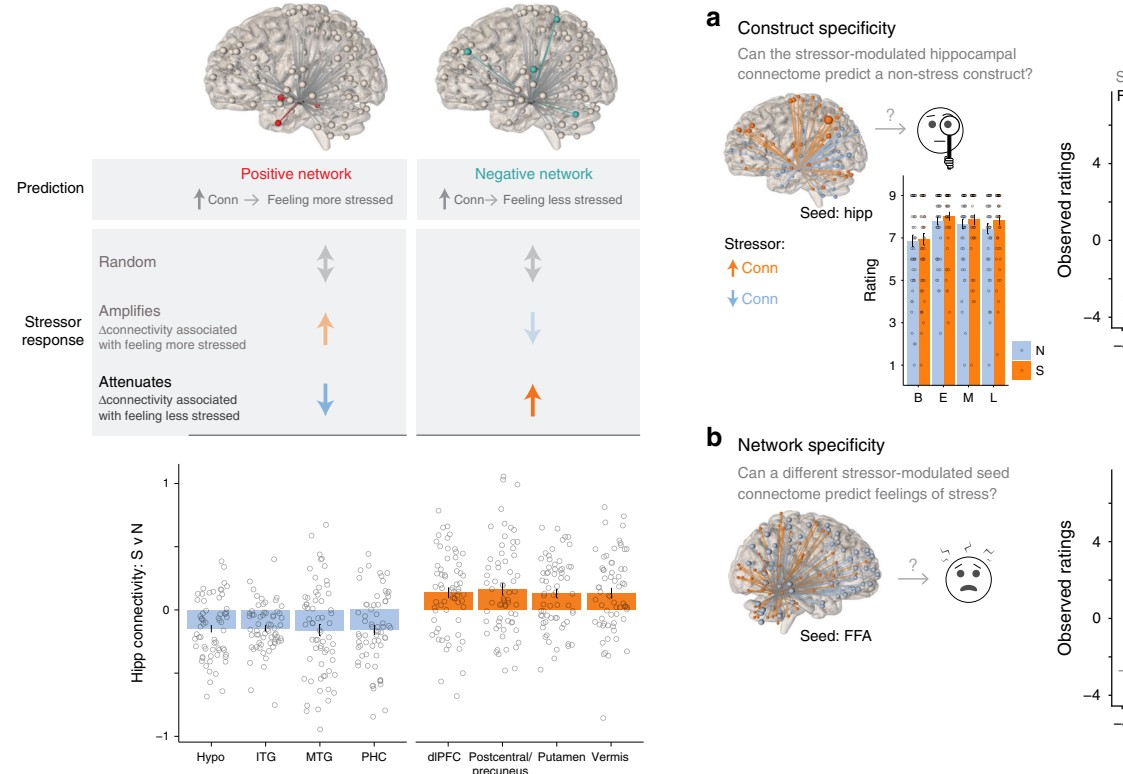

**Fig. 3 Hippocampal networks that predict feelings of stress adaptively respond during stressor exposure.** Predictive hippocampal networks (defined based on ability to predict feelings of stress) could show several distinct response patterns during stressor exposure. These changes could be random, associated with amplifying feelings of stress, or associated with attenuating feelings of stress. Findings showed that, across participants, every network cluster responded consistent with attenuating feelings of stress. Bar graph shows average connectivity for Stressor vs. Neutral, with overlaid dots showing individual participants. Error bars = +1 SE. N = 60 participants. Source data are provided as a Source Data file.

that any stressor-induced changes in functional connectivity could predict feelings of stress. We used a control ROI (FFA, 7 mm-sphere centered on published coordinates[44]) and computed stressor-modulated background connectivity with FFA following the same procedure used for the hippocampal ROIs. This revealed a stressor-responsive FFA network. Notably, this network had even more significant clusters (i.e., model features) than the hippocampal network. We then repeated the sCPM analysis to test whether the FFA network could predict the same dimensions of subjective stress. Supporting the predictive value of the hippocampal network, the FFA network did not successfully predict any of these metrics of subjective stress (example in Fig. 4b). Indeed, all hippocampal models predicting stress significantly out-performed models trained on the FFA network (Supplementary Fig. 2). Interestingly, the FFA network—specifically, connectivity between FFA and clusters in orbitofrontal cortex (OFC)—could successfully predict focus (nonparametric $P = 0.03$; focus vs. stress: $Z = 2.47$, $p = 0.013$). Thus, this analysis revealed a double-dissociation: stressor-modulated hippocampal connectivity predicted feelings of stress (not focus), whereas stressor-modulated FFA connectivity predicted focus (not feelings of stress). Together, these results support the construct and network specificity of the hippocampal models.

Further analyses confirmed that hippocampal model predictions were not confounded by age or motion. Motion, measured

**Fig. 4 Hippocampal models show construct and network specificity.** Representative model is shown here; all models shown in Supplementary Fig. 2. N = 60 participants. Source data are provided as a Source Data file. **a** Construct specificity analysis. The control construct (ratings of focus) was collected at the same time as ratings of stress-related feelings and is shown for baseline, early, mid, and late epochs of Stressor and Neutral conditions (analogous to Fig. 1d). Error bars = +1 SE. Hippocampal networks that successfully predicted stress-related feelings (gray circles) were not able to predict ratings of focus (black triangles) using the same sCPM approach. Correlations near or below zero indicate model failure. **b** Network specificity analysis. The control network (fusiform face area, FFA-seeded connectome) was computed in the same way as the stressor-modulated hippocampal connectome (shown in Fig. 1e). FFA networks did not successfully predict stress-related feelings even when they did predict ratings of focus.

as absolute mean frame-to-frame displacement, did not significantly correlate with the predicted behaviors (rated stress and arousal x Stressor scan motion: both $p > 0.25$; rated stress and arousal x age: all $p > 0.23$). To control for broader motion confounds, we compared model-predicted to observed ratings using partial correlations[43] accounting for motion during these scans (e.g., during both stress images and gray baseline runs) or participant age. All model predictions significantly correlated with observed ratings even with these controls (all $r_s > 0.35$). Finally, in a separate analysis, we confirmed the relevance of the full hippocampal networks (rather than individual clusters) by demonstrating that each network numerically out-performed the single best cluster (Supplementary Table 5).

**Predictive hippocampal networks are consistent.** The above results uncover functional hippocampal connectivity networks that significantly and specifically predict feelings of stress. But how well do these networks capture the feeling of stress? Ideally, hippocampal networks would predict multiple dimensions of stress-related feelings, and extend beyond any particular hippocampal seed or subgroup of participants. We conducted further

# a

Consistency across subjective stress dimensions

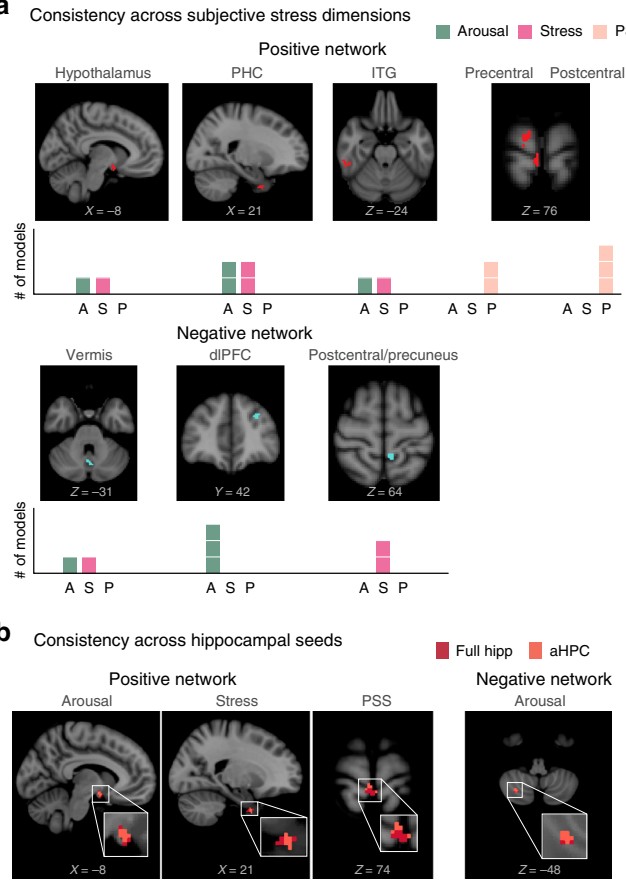

**Fig. 5 Consistency of predictive hippocampal connectivity networks.** N = 60 participants. **a** Clusters identified on every LOO fold as predictive across a significant number of models. Clusters are shown separately for models predicting higher (positive network) and lower (negative network) stress responses. Bar plots indicate the number of models predicting arousal (A), stress (S), and PSS (P) that included that cluster. **b** Overlapping clusters predicting the same stress construct across hippocampal seeds.

analyses to test the generality and consistency of these predictive networks.

As we measured several assays of stress-related feelings (ratings of stress, arousal, and chronic subjective stress [PSS]), and could generate predictions from hippocampal network strength at different timepoints (overall vs. first 2 min of the Stressor scan) with different baselines (relative to the preceding baseline epoch or the Neutral control scan), we tested whether similar clusters were consistently predictive across these analyses (Fig. 5a). In particular, we identified clusters that were selected in every LOO fold of models built to predict each measure of subjective stress (i.e., hippocampal connectivity with these clusters correlated with feelings of stress for every set of N−1 participants). We then used nonparametric permutation testing to assess whether these clusters were selected more often than chance. Connectivity between the hippocampus and hypothalamus (16.7% of models; P = 0.042), PHC (33.3%; P = 0.007), ITG (16.7%; P = 0.03), precentral (16.7%; P = 0.043) and postcentral gyri (25%; P = 0.013) were consistent features of positive networks, in which higher connectivity predicted feeling more stressed. In contrast, connectivity with dlPFC (25%; P = 0.016), cerebellar vermis (16.7%; P = 0.027), and postcentral/precuneus (16.7%: P = 0.047) were consistent features of negative networks. pHPC network clusters did not survive permutation testing (P > 0.05).

Together, these results demonstrate that hippocampal networks can broadly predict the feeling of stress.

We next tested whether these predictive networks were consistent across functional connectivity maps computed using the full hippocampus, aHPC, and pHPC as seed regions. As these connectivity maps were calculated separately, deriving consistent predictive regions across maps would provide further internal validation. For each predictive model, we compared anatomically overlapping clusters identified on every LOO fold using full hippocampus, aHPC, and pHPC as seeds (Fig. 5b; Supplementary Table 2). Notably, both hypothalamus (arousal) and PHC (stress) were identified as part of positive networks, regardless of whether connectivity was computed with the full hippocampus or only the anterior portion.

Finally, we tested whether predictive networks were consistently informative across our participant sample. As men and women have been shown to differ in expressing subjective stress[45,46] and in prevalence of stress-related psychopathology[47,48], it was possible that the identified networks were actually driven by one sex, or were more informative for one sex than the other. In an exploratory analysis, we built seed connectome-based predictive models of interactions to identify hippocampal networks that differentially predicted feelings of stress for men and women (Supplementary Fig. 3). These analyses revealed several pHPC and aHPC networks that predicted opposite feelings of stress and arousal for male and female participants, which included connectivity with dorsomedial and ventromedial PFC. Crucially, none of the regions that predicted overall subjective stress were identified as uniquely predicting stress in men and women. That is, hippocampal connectivity with regions from the positive network (including hypothalamus) and negative network (including dlPFC) did not make different stress predictions for male and female participants. Thus, these findings demonstrate that the predictive hippocampal networks were consistently informative across subgroups of participants.

## Discussion

By developing sCPM, we showed that hippocampal connectivity during a stressor predicts the emergent feeling of stress in novel individuals. We show that stressor exposure leads to widespread changes in functional coupling between the full hippocampus, as well as anterior (aHPC) and posterior (pHPC) subregions, and the rest of the brain. sCPM analyses parsed these patterns into distinct networks that predicted feeling more or less stressed. Identified networks were specific to the construct of stress and the hippocampal connectome, and were consistent across dimensions of subjective stress, hippocampal seeds, and participant subgroups. This work uncovers a role for hippocampal networks in the subjective experience of stress.

The sCPM approach enabled us to find which components of stressor-modulated hippocampal connectivity, an anatomically-specific and hypothesis-driven network, predict the feeling of stress. Crucially, this technique facilitated the discovery of networks relevant to subjective stress by considering all hippocampal connections[35] rather than limiting to specific a priori connections of interest (e.g., amygdala[49], which was not predictive). Instead, we found that hippocampal connectivity with clusters in the hypothalamus, PHC, and ITG formed a positive network (where higher connectivity predicted feeling more stressed), whereas connectivity with dlPFC, postcentral gyrus, and cerebellum formed a negative network, predicting feeling less stressed. Despite the distinct roles of these networks, our findings suggest that individuals engaged both positive and negative networks adaptively to attenuate feelings of stress. That is, participants had higher connectivity with negative networks (whose strength

predicted feeling less stressed), but, at the same time, had lower connectivity with positive networks.

The discovery that hippocampal/hypothalamic coupling predicts the human conscious feeling of stress expands the known role of this circuit in regulating the physiological stress response[14]. This also aligns with recent findings in rodents that anxiety behavior was caused by activating hippocampal neurons which project to the hypothalamus[50]. On average, the hippocampus was negatively correlated with the hypothalamus during the stressor relative to neutral image exposure (possibly consistent with suppression[51] of physiological stress responses by the hippocampus). Having more negative connectivity also predicted feeling less stressed, suggesting similar directionality of physiological and subjective stress responses. Our identified networks also include other regions known to have congruent associations with stress-related physiological and cognitive functions. In the positive network, ITG and PHC BOLD responses correlate with higher blood pressure stress reactivity[52] and predict more intense negative emotions[33]. In the negative network, dlPFC is sensitive to stress hormones and regulates thoughts and attention[53], including diminishing threat responses via cognitive emotion regulation[54]. Together, these findings suggest overlap in higher-level neural networks supporting both subjective and physiological stress[55,56], consistent with recent reports that hippocampal responses could predict both subjective and physiological fear responses[34].

These predictive hippocampal networks were specific to the construct of stress and generalized across multiple facets of the subjective stress response. By capturing individual variation, these predictive models robustly account for a range of subjective stress experiences. Although the hippocampal network was shown to be particularly informative, it is worth noting that other neurobiological mechanisms likely contribute to subjective stress. Widespread plasticity during stress can influence behaviors like coping strategies[28], impulse control[57], and drug craving[58], which may themselves relate to specific aspects of the subjective stress response. Further work is also needed to determine whether this hippocampal network is predictive across stressors that induce varying levels of subjective stress responses. Notably, this analysis approach revealed that a distinct subjective state (focus) could be predicted by a relevant network (FFA/OFC connectivity), consistent with work implicating these regions in attention to salient stimuli[44,59]. These results underscore that the sCPM approach presented here will provide an ideal tool to explore the predictive power of networks centered on other hubs and broadly map neural correlates of subjective states.

In addition to demonstrating the specificity of hippocampal networks, we also showed that they were widely informative across subjective stress measures and subgroups of participants. An exploratory analysis identified distinct hippocampal connectivity networks that predicted opposite feelings of stress for men and women. These included hippocampal connectivity with mPFC regions, associated with emotion regulation[53] and sex-specific stress responses in humans[60] and rodents[61]. While further work is needed to characterize the sex-specific roles of these networks, these results provide an important precedent and technique to discover sex-specific neural predictors of other states.

Finally, by demonstrating that hippocampal connectivity within the first 2 min of stressor exposure can predict subsequent feelings of stress, these results support the prospective power of these models and highlight the importance of early stressor effects. Although many studies of stress actions on human hippocampal function wait over 10 min (for peripheral glucocorticoid elevation), these findings compel investigations of early stress effects on hippocampal cognition[62]. The finding that

hippocampal connectivity could predict future feelings of stress, even when controlling for concurrent feelings, also suggests a causal mechanism in which these networks drive subjective stress responses, although this cannot be confirmed with correlational data. It is also possible that the success of these predictive models is due to a separate factor modulating hippocampal connectivity across individuals (e.g., variability in univariate responses to stimuli) or modulating both brain and subjective stressor responses (e.g., attention to stimuli). Our analyses indicate that these particular factors are unlikely, as background connectivity analyses should mitigate the influence of stimulus-driven univariate responses, and analyses of focus ratings indicate that subjective attention is both uncorrelated with subjective stress and associated with a distinct neural network. Nevertheless, it is important to consider non-causal interpretations of the relationship between hippocampal connectivity and subjective stress.

In conclusion, we show that hippocampal connectivity under stress predicts feelings of stress. sCPM enables the discovery of hypothesis-driven functional connectivity networks that predict clinically relevant behavior. The identified networks provide insight into the neurobiological mechanisms supporting this important subjective component of the stress response, which may in turn have implications for health and psychopathology[2,4].

## Methods

**Participants**. In total, 60 right-handed healthy participants ($N = 31$ female; demographics in Supplementary Table 1) provided written informed consent to participate in the experiment and were included in analyses. This sample size was determined based on prior work showed that significant stressor-modulated brain responses could be observed using this protocol for $N = 30$ participants[28]. To account for sex differences in stress responses[63], we doubled this sample size to have equivalent power for male and female participants. Male and female participants did not differ significantly in age (F: mean 29.68 years [SD 10.05]; M: 29.52 [6.94]; $p > 0.25$) or IQ (F: 113.17 [6.91]; M: 114.48 [7.17]; $p > 0.25$). Participants did not meet any of the following exclusion criteria: current criteria for any moderate/severe substance use disorder; current opiate use; psychosis/severe psychiatric disability; significant medical conditions; regular use of medications that could interfere with the stress response; claustrophobia or ferromagnetic metal in the body (MRI safety); and, for female participants, pregnant or nursing. All participants were light to moderate drinkers per National Institute on Alcohol Abuse and Alcoholism criteria. The Yale Medical School Institutional Review Board approved procedures.

**Procedures**. Participants arrived at the MRI center and had ~30 min to acclimate to the environment. They completed practice trials outside the fMRI scanner, viewing four unique images (not repeated during the experiment) and using the button box to practice making ratings. Immediately prior to the fMRI scan, participants completed guided progressive relaxation (4 min). At 8 AM, the fMRI session began. Participants passively viewed 132 unique images in Stressor (S, 66 images) and Neutral (N, 66 images) conditions using a block design (Fig. 1a, order counterbalanced). There was a recovery period between conditions (~5 min) during which participants were provided progressive relaxation instructions. Each condition contained 8 contiguous runs (66 s each): 2 baseline (5 s gray screen, 1 s inter-stimulus interval [ISI]) followed by 6 image runs (5 s image, 1 s ISI). During each ISI, a black screen was presented with a white central fixation point. Prior to each run, participants were reminded to focus on the presented images. The Stressor condition included highly aversive images including terror, violence, mutilation, fear, and disgust selected from the International Affective Picture System[64] (mean valence rating = 2.34 [SD 0.63], 1: negative/9: positive; arousal = 6.0 [0.83], 1: calm/9: excited), shown to correspond to the discrete emotional categories of anger, disgust, fear, and sadness[65]. Neutral images were identified online based on common neutral/relaxing situations, including images of nature and people reading in a park (valence = 6.07 [4.2], arousal = 3.63 [0.47]). Emotional intensity (valence/arousal ratings) and content (image category) were matched per run within each condition. Stimuli were projected onto a screen that participants viewed using a mirror attached to the head coil. Each image was 880 × 660 pixel resolution (standard for tasks involving IAPS images) and centrally presented, occupying the majority of the screen as shown in Fig. 1b. After every run, participants rated their stress (1: not at all stressed while viewing the pictures, 9: extremely stressed), arousal (1: calm/relaxed, 9: highly aroused or excited, including images from the Self-Assessment Manikin[66]), and focus (1: not at all, 9: very well) using an MRI-compatible button box. Responses were self-paced. Stimuli were presented using E-Prime (2.0)[67]. Following completion of the scan, participants were compensated and went home.

**fMRI parameters**. Scanning was performed using 3 T Siemens MRI systems (Trio and Prisma). Acquisition parameters were the same across scanners with no significant difference in proportion of male and female participants collected using each scanner ($p > 0.25$) or the measures of subjective stress predicted by hippocampal connectivity (all $p > 0.16$). Structural data were acquired using a sagittal high-resolution T1-weighted 3D MPRAGE sequence (2400 ms TR, 1.96 ms TE, flip angle: 8°, FOV: 256 × 256, 208 slices, 1 mm3 isotropic voxels). Functional data were acquired using a multiband (5 slices/RF pulse) gradient EPI sequence (1000 ms TR, 30 ms TE, flip angle: 55°, FOV: 220 × 220, 75 slices, interleaved acquisition, 2 mm³ isotropic voxels; 4 s dummy run pre-acquisition).

**fMRI preprocessing**. Scans were preprocessed using FSL 6.0.1 and AFNI 18.3.08. Data were high-pass filtered at 0.01 Hz to remove low-frequency drifts in signal, and runs with excessive head motion (defined a priori as >1.5 mm absolute mean frame-to-frame displacement, MCFLIRT) were excluded (1 run from 1 participant). A GLM was conducted per run to control for motion and covariates of no interest (FEAT[68]). Regressors included: 6 linear estimated motion parameters, white matter, cerebrospinal fluid, and global mean signal timeseries (each plus temporal derivatives) and stick function regressors for nonlinear motion outliers. To focus on background connectivity, we removed trial-evoked signal (image on/ offset modeled using a boxcar convolved with a double-gamma HRF, plus temporal derivatives).

Model residuals were aligned to a reference functional scan and then to the participant's high-resolution anatomical scan using boundary based registration[69]. Images were warped to MNI space and smoothed to 6 mm FWHM (using 3dBlurToFWHM, which has been shown to help address motion confounds[70]). Data were then bandpass-filtered to leave signal from 0.01-0.1 Hz, a frequency band used to compute background connectivity during extended task blocks[38]. This provided an additional control for task-evoked responses, as stimulus presentation (1 image/6 s; 0.17 Hz) was outside the upper bound of this filter, and helped minimize potential respiration-related artifacts[71]. Smoothed, filtered data were then concatenated into baseline (gray runs 1–2), early (image runs 1–2), mid (image 3–4), and late (image 5–6) epochs (following[28,63]). These epoch boundaries were not signaled to participants during the experiment.

**Regions of interest**. The hippocampus was anatomically defined on each participant's high-resolution MPRAGE scan using FSL's FIRST segmentation. Anterior (aHPC) and posterior (pHPC) hippocampus were defined as the anterior- and posterior-most thirds of this ROI along the longitudinal axis using a custom MATLAB script[72]. Participant-specific ROIs were warped to MNI space for group-level analyses.

**Seed-based connectivity maps**. The average time course of responses within each hippocampal ROI was computed from the preprocessed data (separately per baseline/early/mid/late epochs; 132 s each). ROI timeseries were each correlated with the timeseries of all voxels throughout the brain and resulting $r$ maps were Fisher $z$-transformed. To assess changes induced by stress (and enable us to interpret relative correlation directionality), differences in connectivity during each image epoch (early/mid/late; 528 s total) relative the immediately preceding baseline epoch (B) were computed.

These differential functional connectivity maps were then entered into a second-level linear mixed effects model (3dLME) with Condition, Epoch, and Sex as fixed effects and participant as a random effect. This allowed for changes in connectivity over time based on temporal evolution of feelings of stress and prior reports of changing univariate stress responses[28] and did not assume the same cross-run variability across participants[73]. Importantly, feelings of stress were not included in the model. The contrast used to define significant stressor-modulated clusters, which would be used in the seed connectome-based predictive models, was the Condition contrast of Stressor vs Neutral. To control for multiple comparisons, this contrast was cluster-corrected using the latest 3dClustSim[74]. For voxelwise $p <$ 0.001, we used bi-sided first-nearest neighbor clustering to determine the cluster threshold for $\alpha = .05$. Clusters were labeled using visualization in fsleyes with labels from the Harvard-Oxford Cortical and Subcortical Atlases, Cerebellar Atlas in MNI152 Space, and published coordinates (e.g.,[75]). Networks were visualized using NeuroMArVL (Monash Adaptive Visualization Lab; http://marvl.infotech.monash. edu.au/).

**Seed connectome-based predictive models (sCPM) inputs: brain**. Using significant clusters from ROI seed-based connectivity maps as a mask (full hippocampus: 73 clusters: aHPC: 95; pHPC: 122; FFA [control ROI]: 121), we computed mean hippocampal connectivity per cluster per participant using each participant's z maps. Consistent with 3dLME model inputs, average connectivity per cluster was computed for early/mid/late epochs relative to baseline. To predict stressor-modulated feelings from stressor-modulated hippocampal connectivity, inputs were relative (i.e., connectivity during S-B, or the difference between S-B and N-B).

**sCPM inputs: feelings of stress**. Feelings of stress were assessed every minute for the 2 baseline runs and the 6 Stressor/Neutral condition runs throughout the scans. As with brain connectivity, we used relative feelings of stress to specifically probe stressor-induced responses while accounting for biases in self-reports of subjective stress. Ratings were matched to the brain connectivity inputs (i.e., if the connectivity was S-B, self-reports were also S-B). For prospective models, connectivity from the Early epoch (relative to Baseline) was used to predict feelings of stress during the Late epoch (relative to Baseline).

**sCPM feature selection**. For every LOO loop, clusters (i.e., model features) were selected if they showed above-threshold Spearman correlations with behavior. In most cases, this threshold was $p < 0.05$. If no clusters met this criterion (as when behavior was randomly permutated), we used the top X most predictive clusters determined by rank-ordering coefficients per cluster from strongest to weakest, computing differences between successive coefficients, and selecting all X clusters with correlations above the largest drop in predictive power (i.e., largest difference between successive coefficients).

**sCPM feature consistency**. We ran nonparametric tests to determine whether the same networks were consistently identified across dimensions of subjective feelings of stress. Similar to the nonparametric tests used to assess model predictive power, ratings were randomly shuffled 1000x with all ratings per participant linked per shuffle (i.e., all stress/arousal ratings for subject A were paired with brain connectivity from subject B). All predictive models were generated per iteration (mean arousal, late arousal, etc.) and clusters selected on every LOO fold were stored. This created null distributions of the number of models for which each cluster was selected. $P$ values were computed by comparing the number of models for which each cluster predicted the true behavior to this null distribution.

**Multiple comparisons correction**. Models were trained to predict aspects of subjective stress (arousal, stress, PSS scores) from functional connectivity defined using hippocampal seeds (full, aHPC, pHPC) during Stressor image runs (all six or early only), relative to the immediately preceding baseline or the difference between stress-baseline and neutral-baseline. In other words, per subjective stress measure, 12 models were trained and tested using LOO cross-validation. With strict Bonferroni correction ($0.05/12 = 0.0042$), nonparametric $p$ values from 2/5 models survive. However, as the predictive models in each set are not independent (e.g., maps from full hippocampus/aHPC/pHPC are not independent) this correction is overly strict. To ensure that predictions were robust and reliable, we assessed the consistency of features (functional connections) selected during model training (Fig. 5). If model predictions were driven by noise, we would not expect the same features to be selected in multiple networks. Instead, features were consistently selected across models.

**Reporting summary**. Further information on research design is available in the Nature Research Reporting Summary linked to this article.

## Data availability
The data that support the findings of this study are available from the corresponding author upon reasonable request. The source data underlying Figs. 1d, 2c–d, f–g, i, 3, 4 and Supplementary Figs. 1, 2c–g, 3c–f, i, k are provided as a source data file.

## Code availability
Predictive models were generated using a modified version of publicly available connectome-based predictive modeling (CPM) code (accessible at https://github.com/ YaleMRRC/CPM).

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

## Acknowledgements

This work was supported by NIH grants T32 DA022975, KL2 TR001862 (EVG), R01 AA026844 (DS) and R01 AA013892 (RS).

## Author contributions

E.V.G. and R.S. designed research, R.S. and D.S. performed research, M.D.R. and R.T.C. contributed analytic tools, E.V.G. analyzed data with advice from M.D.R, E.V.G. and R.S. wrote the paper.

## Competing interests

The authors declare no competing interests.

## Additional information



