## [Peer Review File · Nature Communications]

Reviewers' Comments:

Reviewer #1:

Remarks to the authors:

Remarks to the Editor:

This is an interesting article investigating whether functional hippocampal connectivity during acute stress may specifically predict the subjective feeling of stress.

Some questions/thoughts arose during the review:

1) Line 151, Fig. 2: b-d predicted rating of stress (pink), arousal (green) or both (purple)- there is no purple in the graphics.

2) Line 289: How was the sample size calculated?

3) Line 298-308: What kind of images were categorized in Neutral? Were there from the International Affective Picture System? How can you know that they were neutral-relaxing? In this case, aren't they positive instead of neutral?

4) Line 301: The Stress condition contained highly aversive images, therefore, the results of this study are probably limited to stress induced by aversion, and not stress induced by fear or anger, for example, which are more common than induced by aversion. It is important to discuss this limitation. Maybe hippocampal/hypothalamic coupling predicts the feeling of stress only induced by aversion.

Minor:

Line 34: functional neuroimaging (fMRI) - functional magnetic resonance imaging (fMRI)

Reviewer #2:

Remarks to the Author:

Goldfarb and colleagues show that functional connectivity between the hippocampus and other brain regions can be used to infer subjective stress levels. They show that this is not true of other subjective measures that do not relate to stress, nor is it true of the same connectivity analyses seeded in other brain regions.

Overall I found this work to be interesting and timely. Nonetheless, I have some concerns with the current manuscript that make me uncertain whether the data really support the claims. In particular, I am unclear on the functional interpretations that are given to the connectivity results, and more generally I just found it very difficult to understand what was actually done to arrive at the displayed figures in many cases. A complete listing of my concerns is below:

It is almost impossible to interpret the meaning of the reported results without additional information about the task itself, including timing, types of images, and durations. I would recommend that the authors make an additional figure to clarify what participants were doing before, during, and after the functional imaging data was collected.

The authors say that they "regressed out task effects and confounds", but I am unclear on how this is possible. It would be easier to evaluate this if the authors had provided a more clear description (including a figure) on the stress manipulation itself, however my understanding from what has been provided is that participants would see a series of trial unique images in each condition. In all likelihood, some images evoke greater brain responses than other images, and the difference between such pairs should almost certainly be greater for some brain regions in the stressful condition (due to greater salience of images). Furthermore, different image sets could evoke different correlations – for

example, if I use an image set in which all images are either on the left side of the screen or the right then I would likely get a negative correlation between regions in left/right visual cortex – regressing out the response to images won't remove this negative correlation, as half of the images induce one lateralization, and the other half induce the other. Obviously this correlation structure is quite different than what you would get if you presented images centrally, where presumably more positive correlations should be induced across hemispheres of visual cortex. The only way that I can imagine controlling for this type of effect is regressing out the effect of every stimulus – and if images are unique, this would lead to a good deal of overfitting, and likely leave the residual data with more noise than signal. At minimum, I would recommend that the authors provide a more lucid description of their manipulation and clarify what they have done to limit the impact of stimulus-induced brain responses on the regional correlations that they identify. Assuming that my understanding is correct, and it is not possible to remove the possibility that network effects are driven by common responses to trial-unique images, the authors should discuss this as a caveat to their results.

I find the functional claims (eg. "Together, these results suggest an adaptive, multifaceted hippocampal connectivity response that is engaged during stressor exposure to attenuate the feeling of stress") to be a bit far fetched, and as far as I can tell, insufficiently supported by data. First of all, this paradigm doesn't seem particularly well suited to making inferences about what the network might be doing, as it does not involve behavior, other than making occasional self reports about stress and arousal. Second of all, I'm not sure how the evidence presented that precedes this conclusion rules out that the connectivity profile reflects either (1) the feeling of stress or (2) the patterns of input that tend to correspond with it. If it does, and I've missed it, it would be great if a more clear explanation could be provided. Otherwise, it would be reasonable for the claims in the results section to be scaled back to what is supported by available data.

I'm a bit unclear about the logic behind the test of construct specificity. Wouldn't a demonstration of construct specificity require being able to identify a network that changes with respect to focus within subject, and to show that this network predicts changes in focus across subjects? Here the authors seem to use the feature selection set up for stress, choose a weighted readout based on stress, then ask whether that readout predicts focus – I guess I'm just a little bit unclear why this might be expected to be the case. Similarly, did FFA actually show a response to the stress manipulation? It seems that a stronger test of specificity would be to examine one or all regions that respond to the stress manipulation and demonstrate that their functional connectivity does not predict changes in subjective stress.

Do the authors have any way to verify that participants were attending to, or at least looking at the images in the two conditions? It seems that one potential concern would be that individual differences in stress response are reflected by the degree to which participants actually processed (or even saw) the image manipulation. This could go in either direction – which those who process more deeply getting more stressed out (which would not be consistent with data) – or with those who get stressed out rapidly disengaging and no longer attending (which I think would). It would not be surprising if this sort of difference should be visible through connectivity differences, although, in the simplest case where some individuals stop looking altogether, one might expect that such connectivity differences would NOT be selective for a specific region (though see comment above about selectivity).

Minor points:

I'm not sure that I follow this:

"Although hippocampus and hypothalamus were negatively correlated under stress relative to neutral

conditions (possibly consistent with suppression⁴⁹ of physiological stress responses by the hippocampus), individuals who did not show this decrease felt more stressed, suggesting similar directionality of physiological and subjective stress responses.”

Figure 2a should be explained and potentially clarified. For example, I don't see an explanation of what M(+) and M(-) are.

Figure 2: Is connectivity average across selected features/clusters if so, for which LOSO condition?
Figure 2 h & i – what is going on with description – says stress, but shows arousal? Is this important.

Fig 2 – It would be useful if the actual permutation distribution could be displayed for comparison to actual effects.

Why pHPC left out of last figure?

Reviewer #3:

Remarks to the Author:

In this manuscript, Goldfarb and colleagues investigate hippocampal network function during acute stress using machine learning methods. Positive and negative hippocampal stress networks were identified. Connectivity with the positive network (hypothalamus, parahippocampal cortex, inferior temporal gyrus and middle temporal gyrus) predicted higher feelings of stress and connectivity with the negative network (dIPFC, PCG, putamen, cerebellum) predicted lower levels of stress. Human stress confers risk for multiple different physical and psychiatric conditions, therefore understanding the mechanisms underlying stress responses is a critical area for research.

This investigation used a well-validated stress task combined with novel machine learning methods to identify which hippocampal networks are associated with feelings of stress. Other strengths include a relatively large sample size, strong experimental methods, such as controlling study start times, use of both specific hippocampal subregions and time epochs, and additional region and symptom analyses to demonstrate specificity of the findings. This report represents a novel and important contribution to our understanding of how hippocampal network contribute to feelings of stress.

Minor Concerns

1. The participants appear to be generally healthy controls, but defined primarily by absence of a substance use disorder or psychosis. Did any of the participants have current or lifetime history of psychiatric disorders that might relate to either hippocampal alterations or altered stress responses, like PTSD, depression, or anxiety disorders?
2. Multi-band methods have been recently reported by the ABCD project to often have respiration induced artifacts (white stripes in the raw functional images). Can the authors please confirm that the data from this study do not include such artifacts or have had a respiration correction performed?
3. Although a stress task was performed, the trial-evoked signal (image on/offset) was removed to produce a measure of intrinsic or resting state connectivity. Is it possible that removing the trial-evoked signal removed some (or a critical part) of the stress effect? That is, would the results have been similar if the image timing effects weren't removed?

4. Since the reliability of connectivity measures can be related to the amount of imaging time analyzed, it would be helpful to specifically state the amount of minutes/seconds included in the three image epochs.
5. The leave-one out method has some issues, so it was good to see that a 10-fold validation was also performed. The reference to the cross-validation in the main text focuses on the effect sizes. Can the authors please add the p-values for the 10-fold method to supplementary table 1 and also indicate in the main text that while the effect sizes are still in the medium to large range, the 10-fold method did result in lower correlations. This might help the reader to focus on the results that were robust across both methods and therefore most likely to replicate in a future sample.
6. Is there a figure legend for the supplemental figure? It is quite challenging to interpret the figure and to know which brain regions are being represented in the scatterplots.
7. Figure 3 (specificity analyses) might be better in the supplement since these are really just null, confirmatory results. Also, it is a bit confusing to have the figure 3 legend refer to results from Figure 2.
8. In figure 4, it is challenging to understand the inset dots in the context of the figure. Could this information be included in a table instead? It would also be helpful to know the number of models predicting the cluster out of how many models (ie. the percentage).

We appreciate the reviewers' thoughtful comments. All changes to the revised manuscript are indicated in blue font.

Reviewer #1 (Remarks to the Author):

This is an interesting article investigating whether functional hippocampal connectivity during acute stress may specifically predict the subjective feeling of stress.

We appreciate the reviewer's positive comments and careful reading.

Some questions/thoughts arose during the review:

1) Line 151, Fig. 2: b-d predicted rating of stress (pink), arousal (green) or both (purple)- there is no purple in the graphics.

We apologize for the ambiguity. There are purple dots on the brain images (Fig. 2b, e) showing clusters that were identified as predictive across ratings of stress and arousal. We have revised the figure legend to clarify this.

2) Line 289: How was the sample size calculated?

The sample size was determined based on prior work showed that significant stressor-modulated brain responses could be observed using this protocol for N = 30 participants (Sinha et al 2016). To account for sex differences in stress responses (Goldfarb, Seo & Sinha 2019), we doubled this sample size to have equivalent power for male and female participants. This information is now included in the Methods (p. 13).

3) Line 298-308: What kind of images were categorized in Neutral? Were there from the International Affective Picture System? How can you know that they were neutral-relaxing? In this case, aren't they positive instead of neutral?

We agree that further clarification about the experimental design was needed, and have greatly expanded this description on p. 14 (see also Reviewer #2, point 1 below). In brief, these images included photographs of nature, relaxing in a park, etc., and were rated within the neutral range (mean valence rating = 6.07, where 1 = negative and 9 = positive; mean arousal rating = 3.63, where 1 = calm and 9 = excited). As the images were slightly positive and low arousal, we used the term "neutral/relaxing" to describe these stimuli. Crucially, as this study focused on subjective stress, we found that the Neutral condition induced significantly lower ratings of stress and arousal compared to the images in the Stressor condition during all three task epochs (all $p < .001$). These ratings reflect that the stimuli captured the neutral end of the subjective stress spectrum and provide a good control for the Stressor condition.

4) Line 301: The Stress condition contained highly aversive images, therefore, the results of this study are probably limited to stress induced by aversion, and not stress induced by fear or anger, for example, which are more common than induced by aversion. It is important to discuss this limitation. Maybe hippocampal/hypothalamic coupling predicts the feeling of stress only induced by aversion.

We appreciate the reviewer raising this issue and have provided further clarification about the content of the stimuli. In fact, the images included a wide range of threatening,

fear-inducing, and aversive stimuli. The IAPS images used in the current study were previously categorized as generating the discrete emotional categories of anger, disgust, fear, and sadness (Mikels et al 2005), and have added this information to the Methods (p. 13-14). We have also noted this point as a limitation in the discussion (p. 12).

Minor:

Line 34: functional neuroimaging (fMRI) - functional magnetic resonance imaging (fMRI)

We have updated the text accordingly.

Reviewer #2 (Remarks to the Author):

Goldfarb and colleagues show that functional connectivity between the hippocampus and other brain regions can be used to infer subjective stress levels. They show that this is not true of other subjective measures that do not relate to stress, nor is it true of the same connectivity analyses seeded in other brain regions.

Overall I found this work to be interesting and timely. Nonetheless, I have some concerns with the current manuscript that make me uncertain whether the data really support the claims. In particular, I am unclear on the functional interpretations that are given to the connectivity results, and more generally I just found it very difficult to understand what was actually done to arrive at the displayed figures in many cases. A complete listing of my concerns is below:

We appreciate the reviewer's positive feedback and constructive suggestions. Across these comments, we noted that there was some ambiguity between the stressor (the image viewing condition) and the feeling of stress (what was predicted by the hippocampal network), and have revised the text and figures to help clarify.

1. It is almost impossible to interpret the meaning of the reported results without additional information about the task itself, including timing, types of images, and durations. I would recommend that the authors make an additional figure to clarify what participants were doing before, during, and after the functional imaging data was collected.

We agree that further details about the task design would be useful and have updated the description of the methods (p. 13-14) and Fig. 1a-d accordingly (reproduced as Fig. R1 below).

Fig. R1 Stressor exposure influences feelings of stress and hippocampal connectivity. **a** Design. Functional neuroimaging data (fMRI) were acquired as participants were exposed to 6 consecutive runs of neutral/relaxing (Neutral condition, N) or highly aversive images (Stressor, S) immediately preceded by fixation (Baseline, 2 runs). **b** In each condition, images (5s) alternated with interstimulus intervals (1s) during six 66-sec runs. **c** Ratings of feelings of stress were collected after each run on 9-point visual analog scales. **d** The stressor condition led to significant sustained increases in ratings of arousal and stress. B = baseline, E = early, m = Middle, L = late (as in a). Error bars = ± 1 SE, overlaid dots show individual participants.

2. The authors say that they “regressed out task effects and confounds”, but I am unclear on how this is possible. It would be easier to evaluate this if the authors had provided a more clear description (including a figure) on the stress manipulation itself, however my understanding from what has been provided is that participants would see a series of trial unique images in each condition. In all likelihood, some images evoke greater brain responses than other images, and the difference between such pairs should almost certainly be greater for some brain regions in the stressful condition (due to greater salience of images). Furthermore, different image sets could evoke different correlations – for example, if I use an image set in which all images are either on the left side of the screen or the right then I would likely get a negative correlation between regions in left/right visual cortex – regressing out the response to images won’t remove this negative correlation, as half of the images induce one lateralization, and the other half induce the other. Obviously this correlation structure is quite different than what you would get if you presented images centrally, where presumably more positive correlations should be induced across hemispheres of visual cortex. The only way that I can imagine controlling for this type of effect is regressing out the effect of every stimulus – and if images are unique, this would lead to a good deal of overfitting, and likely leave the residual data with more noise than signal. At minimum, I would recommend that the authors provide a more lucid description of their manipulation and clarify what they have done to limit the impact of stimulus-induced brain responses on the regional correlations that they identify. Assuming that my understanding is correct, and it is not possible to remove the possibility that network effects are driven by common responses to trial-unique images, the authors should discuss this as a caveat to their results.

We apologize for the ambiguity in this statement. To focus on intrinsic background connectivity, we regressed out stimulus-evoked responses using a single regressor of a boxcar for image on/offset (5s image, 1s inter-stimulus interval) convolved with a

hemodynamic response function and its temporal derivatives. To further ensure that changes in connectivity were not simply a function of this regular stimulus periodicity, we took the residuals and applied a bandpass filter (0.01-0.1 Hz) at a range below the stimulus frequency (1 image every 6 seconds = 0.17 Hz). This approach was also employed by Tompary, Duncan & Davachi (2015). As a further control against variability in univariate responses, we regressed out global mean signal (computed per each 66 sec run). Additional noise confounds included average white matter and CSF timeseries, and both linear and nonlinear motion regressors. We have revised the text to clarify this point and explain that this approach was meant to account for synchronized stimulus-evoked responses (p. 4).

To investigate the potential contribution of task effects, we tested the synchrony of responses across participants. As the same task structure was used for each participant, evoked responses should be synchronized across participants, whereas intrinsic activity is idiosyncratic and should not be synchronized (see Tompary, Al-Aidroos & Turk-Browne, 2018). To test response synchrony, we extracted each participant's mean hippocampal timeseries after the processing steps described above (i.e., the same timeseries that was used to determine stressor-modulated whole-brain hippocampal connectivity). For each run, we correlated the hippocampal timeseries from each participant with the average hippocampal timeseries from all other participants. As shown in Fig. R2 below, this revealed that there were no significant correlations across participants in any run, supporting the conclusion that these patterns were not driven by task-evoked responses.

Fig. R2. Lack of cross-subject synchrony in hippocampal timeseries. As each participant experienced the same task structure, evoked responses would likely be synchronized across participants, but intrinsic activity would be idiosyncratic and not synchronized. We did not observe significant cross-subject synchrony in any run of the experiment. Boxplots show median, 25th and 75th percentiles (hinges) and 1.5*IQR (whiskers) from correlating each subject with the average timeseries of all other subjects from that run.

While we agree with the reviewer that this approach cannot account for unique features of a single stimulus, it is not clear how this would influence the predictive results presented in the manuscript. Each participant viewed the same set of stimuli (in different orders) but had unique profiles of emotional stress responses (which were measured across sets of 22-66 images). Thus, the effects of individual stimuli on hippocampal connectivity would have to systematically differ across individuals in a way that matched these aggregated stress ratings in order to influence the predictive models.

Finally, as requested in comment #1, we have also provided further details about the image stimuli in the Methods (p. 13). In brief, participants viewed 132 unique images (66 during the Stressor and 66 during the Neutral condition). Each image occupied the entire screen as shown in Fig. R1b above.

3. I find the functional claims (eg. “Together, these results suggest an adaptive, multifaceted hippocampal connectivity response that is engaged during stressor exposure to attenuate the feeling of stress”) to be a bit far fetched, and as far as I can tell, insufficiently supported by data. First of all, this paradigm doesn’t seem particularly well suited to making inferences about what the network might be doing, as it does not involve behavior, other than making occasional self reports about stress and arousal. Second of all, I’m not sure how the evidence presented that precedes this conclusion rules out that the connectivity profile reflects either (1) the feeling of stress or (2) the patterns of input that tend to correspond with it. If it does, and I’ve missed it, it would be great if a more clear explanation could be provided. Otherwise, it would be reasonable for the claims in the results section to be scaled back to what is supported by available data.

We appreciate the reviewer’s comment and agree that these points require clarification and further support.

We have updated the analyses supporting our interpretation that the hippocampal response was adaptive and associated with attenuated feelings of stress (p. 7). In brief, all hippocampal connections that changed during stressor exposure were considered as potentially predicting the feeling of stress. That is, the model was not told whether stressor exposure *increased* or *decreased* the strength of each functional connection.

After using the model to identify networks that predicted feeling stressed, we next asked how stressor exposure modulated these networks. There were several possibilities, including: 1) Random changes (stressor both increased and decreased functional connections within predictive networks); 2) Amplification (stressor changed connectivity within each network such that feelings of stress would increase); and 3) Attenuation (stressor changed connectivity within each network such that feelings of stress would decrease).

Strikingly, we found that the average response to the stressor – across every single network cluster – corresponded to attenuated feelings of stress. In other words, the stressor did not have an overall effect on hippocampal functional connectivity. Instead, it differentially affected hippocampal functional connections depending on whether they predicted more or less subjective stress. Specifically, hippocampal connections that predicted *increased* stress *decreased* in strength in response to the stressor, whereas hippocampal connections that predicted *decreased* stress *increased* in strength in response to the stressor.

We have created a new figure to illustrate this analysis (Fig. 3, reproduced as Fig. R3 below; also see Supplementary Table 4) and revised the discussion in the text (p. 7). We believe this pattern supports the conclusion that these networks, on average, were engaged in an adaptive fashion such that feelings of stress would be attenuated. As this pattern extended across both positive and negative predictive networks, we consider this to be a multifaceted response.

Fig. R3. Hippocampal networks that predict feelings of stress adaptively respond during stressor exposure. Predictive networks (defined based on ability to predict feelings of stress) could show several distinct response patterns during stressor exposure. These changes could be random, associated with amplifying feelings of stress, and associated with attenuating feelings of stress. Across participant, every network cluster responded in such a way that feelings of stress would be attenuated. Bar graph shows mean and SEM of Z scores for each volume cluster (identified in the Stressor > Neutral contrast).

With regard to interpreting the functional connectivity profile: the seed-based connectivity map (shown in Fig. 1) was computed as clusters that showed statistically different patterns of connectivity with the hippocampus during Stressor relative to Neutral conditions. The feeling of stress was not part of this computation. Thus, we agree with the reviewer that this profile could reflect the feeling of stress and/or the input that corresponds with it. To determine which components of this profile specifically corresponded to the feeling of stress, we used the seed connectome-based predictive modeling (sCPM) machine learning approach. Using this approach, we could isolate which components of the stressor-modulated profile were associated with individual differences in the feeling of stress. We further decomposed this profile into networks that predicted increases (e.g., hippocampus/hypothalamus connectivity) and decreases (e.g., hippocampus/dIPFC connectivity) in the feeling of stress. Notably, there were components of the stressor-modulated connectivity profile (e.g., hippocampus/amygdala

connectivity) that did not predict the feeling of stress, demonstrating that the entire connectivity profile was not predictive. We have revised the text to clarify this point (p. 5).

In addition, one of the exciting parts of these analyses is that some identified networks were truly prospective: that is, connectivity within the first ~2 minutes of stressor exposure predicted feelings of stress measured ~2 minutes later. These analyses thus demonstrate that hippocampal connectivity temporally preceded feelings of stress and suggest that connectivity within predictive networks gave rise to these feelings. However, it is possible that these networks were more strongly linked to feelings that were measured concurrently (suggesting that feelings might come first). As an exploratory analysis, we tested whether network strength was associated with subsequent feelings of stress, even when accounting for concurrent feelings of stress. Of the four networks that predicted subsequent stress, three continued to show significant associations with subsequent stress even with this control, (hipp: Late stress[+]: $r = .42$, $p < .001$; Late arousal[-]: $r = -.51$, $p < .001$; pHPC: Late arousal[-]: $r = -.28$, $p = .03$). These analyses further support the idea that hippocampal networks adapted during the stressor to influence subsequent subjective stress and have been incorporated into the text (p. 7-8).

Finally, the goal of the task design was to measure stressor-induced changes in hippocampal connectivity while inducing a subjective stress response. Participants were instructed to focus on the pictures and provide self-reports of their subjective state approximately once every minute (8 ratings each for Stressor and Neutral conditions). This frequent monitoring allowed us to capture fluctuations in emotional state that could be predicted using concurrent hippocampal connectivity. We have updated the task design figure (Fig. 1, as requested in #1 above) to clarify these points.

4. I'm a bit unclear about the logic behind the test of construct specificity. Wouldn't a demonstration of construct specificity require being able to identify a network that changes with respect to focus within subject, and to show that this network predicts changes in focus across subjects? Here the authors seem to use the feature selection set up for stress, choose a weighted readout based on stress, then ask whether that readout predicts focus – I guess I'm just a little bit unclear why this might be expected to be the case.

The goal of the construct specificity analysis was to test whether stressor-induced changes in hippocampal connectivity would predict any concurrent subjective state, or if they were specific to the feeling of stress. We tested this in two ways.

First, we took the stressor-modulated hippocampal network and used a new feature selection process to see whether any features of this network could predict focus. The feeling of stress was not part of the network definition or feature selection. This analysis (Fig. 4, reproduced as Fig. R4 below, as well as Supplementary Fig. 2) demonstrated that, although features of this hippocampal network could predict the feeling of stress, the network could not successfully predict another concurrent subjective state. We have conducted additional Z tests demonstrating that predictions of stress are stronger than predictions of focus (p. 8, Supplementary Fig. 2).

Second, we tested whether the network features that did predict the feeling of stress would show any correlation with the feeling of focus. We agree with the reviewer that the

logic of this second analysis was less compelling (although it does partially speak to the concern about attention raised in #6), and have removed it.

5. Similarly, did FFA actually show a response to the stress manipulation? It seems that a stronger test of specificity would be to examine one or all regions that respond to the stress manipulation and demonstrate that their functional connectivity does not predict changes in subjective stress.

The FFA network was defined as regions that showed a significant change in connectivity with FFA during the Stressor compared to the Neutral condition (voxelwise $p < .001$, cluster corrected $\alpha = .05$). This is the same procedure that was employed to identify the stressor-modulated hippocampal network. Consistent with the reviewer's comment, this analysis indeed demonstrated that functional connectivity in a stressor-responsive network did not predict changes in subjective stress. We also conducted Z tests demonstrating that hippocampal networks significantly outperformed FFA networks for all predictive models (revised Supplementary Fig. 2). We have revised the description in the manuscript to clarify this analysis.

We also note that, from a machine learning perspective, the stressor-modulated FFA network served as an ideal control condition to test for network specificity. This network had 121 significant stressor-modulated connections (model features) that we could use for prediction, whereas the hippocampal network only had 73 – if this was simply a question of more model features leading to more successful predictions, the FFA network should have done a better job than the hippocampal network of predicting subjective stress.

Inspired by the above comments #4 and #5, we took a more critical look at our specificity analyses. We considered the possibilities that: 1) hippocampal connectivity could not predict “focus” because (for some reason) these ratings were not well-suited to reliable predictions, and 2) the FFA network (for some reason) could not generate reliable predictions. If either of these were true, it would undermine the conclusions that we drew from the specificity analyses.

To test these possibilities, we examined whether the stressor-modulated FFA connectivity could predict ratings of focus. We found that FFA connectivity with clusters in orbitofrontal cortex could successfully predict focus ratings. This demonstrated that both 1) and 2) above were not supported, and further validated our specificity analyses. It also provided a novel double-dissociation: stressor-modulated hippocampal connectivity predicted feelings of stress (but not focus), and stressor-modulated FFA connectivity predicted focus (but not feelings of stress). Furthermore, this finding demonstrates the utility of the sCPM approach for understanding the functional roles of different neural networks and broadly mapping multiple subjective states. This new finding has been incorporated into the manuscript (p. 8-9, 12; Fig. 4, reproduced below as Fig. R4).

Fig. R4. Hippocampal models show construct and network specificity. **a** Construct specificity analysis. The control construct (ratings of focus) was collected at the same time as ratings of stress-related feelings and is shown for baseline, early, mid, and late epochs of Stressor and Neutral conditions (analogous to Fig. 1d). Hippocampal networks that successfully predicted stress-related feelings (gray circles) were not able to predict ratings of focus (black triangles) using the same sCPM approach. Correlations near or below zero indicate model failure. Representative model shown here; all models shown in Supplementary Fig. 2a.

b Network specificity analysis. The control network (fusiform face area [FFA]-seeded connectome) was computed in the same way as the stressor-modulated hippocampal connectome (shown in Fig. 1e). FFA networks did not successfully predict stress-related feelings even when they did predict ratings of focus. FFA-seeded versions of all predictive hippocampal models shown in Supplementary Fig. 2b.

6. Do the authors have any way to verify that participants were attending to, or at least looking at the images in the two conditions? It seems that one potential concern would be that individual differences in stress response are reflected by the degree to which participants actually processed (or even saw) the image manipulation. This could go in either direction – which those who process more deeply getting more stressed out (which would not be consistent with data) – or with those who get stressed out rapidly disengaging and no longer attending (which I think would). It would not be surprising if this sort of difference should be visible through connectivity differences, although, in the simplest case where some individuals stop looking altogether, one might expect that such connectivity differences would NOT be selective for a specific region (though see comment above about selectivity).

We appreciate the reviewer raising this issue and agree that participants' levels of attention could influence both stressor-induced feelings of stress and stressor-modulated hippocampal connectivity patterns. To address this issue, we obtained self-reports of focus. As noted above, focus ratings were predicted by FFA/OFC connectivity, regions which have been associated with attention to salient information (e.g., Rosenberg et al 2015 *Neuroimage*; Hunt et al 2018 *Nat Neurosci*), providing support for using this as an index of attention. We also examined head motion during the scan, as a possible proxy for avoiding looking at the images (which occupied most of the screen).

Overall, participants self-reported high levels of focus which were not different for the stress and neutral images (paired t-tests assessing condition differences were not significant) and did not show substantial head motion in either condition. This is now included on p. 8. We ran a series of analyses examining these attention-related indices, which are summarized in Table R1 below. Spearman's correlations showed that

individual differences in focus or motion did not systematically vary with either feelings of stress or connectivity in predictive hippocampal networks. This is also consistent with our earlier finding that the strength of connectivity in stress-predictive networks did not correlate with subjective ratings of focus (see response to #4 above).

Finally, one concern would be that our ability to predict feelings of stress was in some way influenced by individual differences in attention. To test this, we took the absolute difference between predicted and observed stress scores for each participant in each model, and tested whether these values correlated with focus or motion. We found no statistically significant associations between attention indices and model success (especially if accounting for multiple hypothesis testing).

Together, these results suggest that participants were attending to both neutral and stressor stimuli, and that variability in attention did not explain individual differences in subjective stress response, hippocampal connectivity, or predictive model performance.

Table R1. Proxies for attention do not explain responses to stressor or predictive model

Difference between conditions					
	Stressor		Neutral		Difference?
	M (SD)		M (SD)		
Focus (rating)	7.91 (1.56)		7.63 (1.61)		p = .13
Motion (mm)	0.22 (0.06)		0.217 (0.08)		p = .4
Correlation with subjective stress					
	Ratings of stress		Ratings of arousal		
	r_s (p-value)		r_s (p-value)		
Focus	-.02 (.86)		.12 (.37)		
Motion	-.24 (.07)		-0.16 (.23)		
Correlation with stressor-induced hippocampal connectivity					
	Positive network		Negative network		
Focus	-.03 (.8)		-.03 (.81)		
Motion	.05 (.69)		.09 (.48)		
Correlation with predictive model success (abs[Observed – Predicted])					
	Late stress (+)	Late stress (-)	Mean arousal (+)	Late arousal (-)	pHPC: Late arousal (-)
Focus	-.02 (.86)	.02 (.9)	-.04 (.76)	-.18 (.17)	-.07 (.58)
Motion	-.11 (.39)	-.24 (.06)	.08 (.57)	-.12 (.34)	-.12 (.35)

performance

Minor points:

7. *I'm not sure that I follow this:*

“Although hippocampus and hypothalamus were negatively correlated under stress relative to neutral conditions (possibly consistent with suppression⁴⁹ of physiological stress responses by the hippocampus), individuals who did not show this decrease felt more stressed, suggesting similar directionality of physiological and subjective stress responses.”

We apologize for the ambiguity and have revised this statement as follows (p. 12):

On average, the hippocampus was negatively correlated with the hypothalamus during the stressor relative to neutral image exposure (possibly consistent with suppression⁴⁹ of physiological stress responses by the hippocampus). Having more negative connectivity also predicted feeling less stressed, suggesting similar directionality of physiological and subjective stress responses.

We also note that the new Figure 3 (shown above as Fig. R3) should help clarify the dissociation between the direction of predictive networks and how connectivity changed during the stressor.

8. Figure 2a should be explained and potentially clarified. For example, I don't see an explanation of what M(+) and M(-) are.

We have revised the schematic of the analyses (new Fig. 2a, reproduced as Fig. R5 below) to help clarify this point. In brief, M(+) referred to mean hippocampal connectivity with the clusters identified as belonging to the positive network (i.e., more connectivity = higher feelings of stress), whereas M(-) referred to mean hippocampal connectivity with negative network clusters (i.e., more connectivity = lower feelings of stress). This is now depicted visually and explained in further detail in the figure legend.

Fig. R5. Stressor-induced hippocampal connectivity networks predict feelings of stress. **a** Schematic of seed connectome-based predictive model (sCPM) analysis and summary of predictive networks. The goal of this analysis was to determine whether stressor-modulated hippocampal connectivity could predict feelings of stress and, if so, to identify the clusters that were part of positive (predict feeling more stressed) and negative (predict feeling less stressed) networks. We used leave-one-out cross-validation to determine which clusters correlated (positively and negatively) with feelings of stress across N-1 participants, and used these linear models to predict feelings of stress in left-out participant N. **b-g** Hippocampal functional connectivity networks predict feelings of stress. **b-d** Positive networks separated by whether they predicted ratings of stress (pink), arousal (green) or both (purple). **b** Anatomical distribution of positive network. **c-d** Summary of network performance. Left, predictive power; Spearman's correlation (r_s) of model-predicted with observed ratings. Histogram shows distribution of correlation values from 1000 iterations of randomly-shuffled feelings of stress used to nonparametrically determine P values. Right, correlation between overlap network strength (i.e., mean of clusters selected on every leave-one-out iteration) and observed ratings. **e-g** Negative networks (predicting lower feelings of stress). **h,i** Posterior hippocampal functional connectivity network predicting lower arousal

9. *Figure 2: Is connectivity average across selected features/clusters if so, for which LOSO condition?*

We appreciate the reviewer's careful reading. In Fig. 2, the predictive power (e.g., the left plot in Fig. 2c with black dots) shows the predictions derived from every LOSO loop plotted against the true values. The relationship between brain and behavior (e.g., the right plot in Fig. 2c with pink dots) shows mean functional connectivity in the "overlap networks"; that is, the clusters that were selected on every single LOSO iteration. We have revised the legend to clarify this (p. 6).

10. *Figure 2 h & i – what is going on with description – says stress, but shows arousal? Is this important.*

We apologize for the confusion. As emphasized in the updated Fig. 1c, we quantified "feelings of stress" as subjective ratings of arousal and stress. In Fig. 2, we separated the predictive models based on whether they predicted ratings of stress (shown in pink; Fig. 2c,f) and arousal (green; Fig. 2d,g,i). Importantly, several of these network components were identified as predictive across subjective ratings of arousal, stress, and chronic stress scores (see Fig. 5), but our aim in Fig. 2 was to clearly demonstrate the predictions of each model. We have revised the figure legend to clarify this (p. 6, see above).

11. *Fig 2 – It would be useful if the actual permutation distribution could be displayed for comparison to actual effects.*

We agree and have now incorporated these distributions into Fig. 2 (reproduced as Fig. R5 above).

12. *Why pHPC left out of last figure?*

We appreciate the reviewer's detailed analysis of this figure. For Fig. 4a, none of the predictive pHPC clusters were identified in a statistically significant number of models (middle frontal gyrus: $P = .09$, cerebellum: $P = .11$, precentral gyrus: $P = .12$), and we have added this information to the text (p. 10). For Fig. 4b, the pHPC-seed connectivity map did not contain clusters that overlapped with the predictive clusters from the hippocampus-seed connectivity map. To help clarify this (and consistent with Reviewer #3's request for the information from this figure to be presented in tabular format), we have added a new Supplementary Table 2 showing the MNI coordinates of all overlap clusters (i.e., clusters that were identified as statistically significant using full hippocampus, aHPC, and pHPC as seeds, and also showed ≥ 1 voxel overlap across seed maps).

Reviewer #3 (Remarks to the Author):

In this manuscript, Goldfarb and colleagues investigate hippocampal network function during acute stress using machine learning methods. Positive and negative hippocampal stress networks were identified. Connectivity with the positive network (hypothalamus,

parahippocampal cortex, inferior temporal gyrus and middle temporal gyrus) predicted higher feelings of stress and connectivity with the negative network (dlPFC, PCG, putamen, cerebellum) predicted lower levels of stress. Human stress confers risk for multiple different physical and psychiatric conditions, therefore understanding the mechanisms underlying stress responses is a critical area for research.

This investigation used a well-validated stress task combined with novel machine learning methods to identify which hippocampal networks are associated with feelings of stress. Other strengths include a relatively large sample size, strong experimental methods, such as controlling study start times, use of both specific hippocampal subregions and time epochs, and additional region and symptom analyses to demonstrate specificity of the findings. This report represents a novel and important contribution to our understanding of how hippocampal network contribute to feelings of stress.

We appreciate the reviewer’s positive feedback and helpful comments.

Minor Concerns

1. The participants appear to be generally healthy controls, but defined primarily by absence of a substance use disorder or psychosis. Did any of the participants have current or lifetime history of psychiatric disorders that might relate to either hippocampal alterations or altered stress responses, like PTSD, depression, or anxiety disorders?

We agree that this is an important question. Of our community sample of 60 participants, 16 (26.7%) met current and/or former criteria for depression, anxiety disorders, or posttraumatic stress disorder. This information is summarized in the new Supplementary Table 1 (reproduced below, Table R2).

Table R2. Participant demographics

		Number (%) participants
Demographics		
Gender		
Age		
Psychiatric History		
Any depression, anxiety, PTSD (current or lifetime)		16 (26.7%)
Depression		
	Past month	1 (1.7%)
	Lifetime	10 (16.7%)
Anxiety Disorder		
	Past 6 months	5 (8.3%)
	Lifetime	7 (10.6%)
PTSD		
	Past 6 months	5 (8.3%)
	Lifetime	6 (10%)

We next examined whether these individuals showed significant differences in stressor-induced feelings of stress or hippocampal connectivity (new Supplementary Fig 1, Fig. R6 below) relative to those without this history. There were no significant differences between these groups in ratings of stress-related feelings (arousal: $t_{58} = .92, p = .36$; stress: $t_{58} = .86, p = .39$). Likewise, there were generally no statistically significant differences in stressor-modulated hippocampal connectivity with clusters from predictive

networks (positive – ITG: $t_{58} = -.97$, $p = .34$; PHC: $t_{58} = -.43$, $p = .67$; MTG: $t_{58} = 1.48$, $p = .14$; negative – dlPFC: $t_{58} = -1.22$, $p = .23$; vermis: $t_{58} = -.28$, $p = .77$; putamen: $t_{58} = .46$, $p = .65$; postcentral/precuneus: $t_{58} = .1$, $p = .92$). However, there was a borderline difference in hippocampus/hypothalamus connectivity ($t_{58} = 2.01$, $p = .049$), with individuals without psychiatric history showing significantly lower connectivity during the stressor. As having more negative hippocampus/hypothalamus connectivity appears to be adaptive for attenuating feelings of stress (Fig. R3), this finding suggests an intriguing mechanism by which patients with those with psychiatric history may show difficulties in adapting during stressor exposure. Although the current study was not designed to address this question (and this difference between groups would not remain statistically significant after accounting for multiple comparisons across all network clusters), we have mentioned these post-hoc analyses in the text as they suggest an important direction for future research (p. 8).

Fig. R6. Stressor responses vary with psychiatric history. **a** Participants meeting either current or lifetime criteria for anxiety, depression, or posttraumatic stress disorder (PTSD; SCID-5 criteria; shown in turquoise) did not self-report significantly higher ratings of stress or arousal during the stressor (S) relative to the neutral condition (N) compared to participants without these diagnoses (gray). **b** Overall, stressor-modulated hippocampal connectivity with predictive clusters did not significantly differ with psychiatric history. However, participants with mood or anxiety disorder history did have significantly higher hippocampal/hypothalamus connectivity, suggesting a less adaptive response during the stressor.

2. Multi-band methods have been recently reported by the ABCD project to often have respiration induced artifacts (white stripes in the raw functional images). Can the authors please confirm that the data from this study do not include such artifacts or have had a respiration correction performed?

We appreciate the reviewer calling attention to this issue. We visually inspected a subset of runs (from 10 randomly selected participants) and did not observe these artifacts.

In addition, a recent report by Fair and colleagues (2020), which also highlights respiratory-related artifacts in single band fMRI data, suggests that the current analyses avoid some of the concerns associated with respiration. First, they raise the concern that respiration can generate apparent head motion (not associated with fMRI data quality), which would lead researchers to erroneously discard data. We only discarded 2 runs (132s total) based on above-threshold head motion. Second, they note that respiration-

related signal should peak at 0.2-0.3 Hz (which should be preserved given the 1s TR in the current design), and our data were bandpass-filtered at a lower range (0.01-0.1 Hz). Indeed, they suggest that such low-pass filters may help ameliorate these artifacts. We have now included these aspects as minimizing any possible respiration-related artifacts in the Methods section (p. 14).

3. Although a stress task was performed, the trial-evoked signal (image on/offset) was removed to produce a measure of intrinsic or resting state connectivity. Is it possible that removing the trial-evoked signal removed some (or a critical part) of the stress effect? That is, would the results have been similar if the image timing effects weren't removed?

We repeated the analyses of the fMRI data for every participant using the same preprocessing steps described in the text (e.g., motion correction, regressing out CSF/WM/global signal, bandpass filtering residuals, etc.) but omitting the task regressor. As shown below, group-level results for Stressor > Neutral were very similar to those generated when the task regressor was included (Fig. R7).

Fig. R7. Effects of regressing out trial-evoked responses on stressor-modulated hippocampal connectivity. Each participant's data was preprocessed using the pipeline described in the main text, but one version (group-level results shown at left) included a regressor for the task, while the other version (shown at right) did not. Specifically, regressing out task responses involved a boxcar for stimulus on/offset convolved with a double-gamma HRF plus temporal derivatives. As illustrated here, there was a high degree of overlap between networks. Warm colors = Stressor > Neutral condition; Cool colors = Neutral > Stressor. Voxels visualized at $p < .01$, uncorrected.

Next, we identified clusters that were statistically significant from this new model at $p < .001$ and $\alpha = .05$, leading to 96 clusters (compared to 73 clusters when task was regressed out). Below we show overlaps of these maps, highlighting clusters that were identified in our predictive models (Fig. R8). In brief, although most of these predictive clusters were identified as having significant stressor-modulated hippocampal connectivity using both analysis approaches (including hypothalamus, parahippocampal cortex, inferior temporal gyrus, and dIPFC), some clusters were not (e.g., cerebellar vermis and middle temporal gyrus).

Fig. R8. Effects of regressing out trial-evoked responses on stressor-modulated hippocampal connectivity with predictive networks. Top panel: Clusters showing significantly higher connectivity during Stressor relative to Neutral condition; bottom panel: clusters showing significantly lower connectivity during Stressor relative to Neutral. Clusters defined after regressing out task (original version) are shown in magenta; clusters defined without regressing out task shown in yellow; overlap shown in orange. Both sets of maps are defined at voxelwise $p < .001$, $\alpha = .05$. Clusters identified as predictive of feelings of stress using seed connectome-based predictive modeling are highlighted in white squares.

As the goal of our analyses was to examine whether intrinsic, idiosyncratic fluctuations in hippocampal connectivity were associated with feelings of stress (see response to Reviewer 2, #2 above), we believe the predictions generated using the “background connectivity” maps (which regress out task on/offset) provide the optimal approach to address this question.

4. Since the reliability of connectivity measures can be related to the amount of imaging time analyzed, it would be helpful to specifically state the amount of minutes/seconds included in the three image epochs.

We appreciate the reviewer’s suggestion and have added this to the methods (p. 13-14). For reference, we computed connectivity separately per 132-sec epochs and conducted group-level analyses of connectivity (Stressor vs Neutral) using three image epochs (528

sec total) relative to a baseline (fixation) epoch (132 sec). While we agree that the reliability of connectivity measures, especially during rest, can be influenced by the duration of imaging time analyzed, we also note that successful prediction has been performed using functional connectivity during task epochs as short as 22.5 sec (Gonzalez-Castillo et al 2015 *PNAS*) and rest epochs as short as 30 sec (Rosenberg et al 2020 *PNAS*).

5. The leave-one out method has some issues, so it was good to see that a 10-fold validation was also performed. The reference to the cross-validation in the main text focuses on the effect sizes. Can the authors please add the *p*-values for the 10-fold method to supplementary table 1 and also indicate in the main text that while the effect sizes are still in the medium to large range, the 10-fold method did result in lower correlations. This might help the reader to focus on the results that were robust across both methods and therefore most likely to replicate in a future sample.

We have added this statement to the main text (p. 5). We estimated *p* values using Kolmogorov-Smirnov (KS) tests (as in Rudolph et al 2018 *Nat Neurosci*) and added them to Supplementary Table 3. However, consistent with the discussion by Rudolph and colleagues, we would recommend that the emphasis for robustness and replication should be on the effect sizes.

For reference, we have plotted the results of 10-fold cross validation for each model below (Fig. R9).

Fig. R9. Results of *k*-fold cross validation. To account for potential bias due to the leave-one-out (LOO) cross-validation procedure, we implemented a 10-fold cross-validation approach. For each model, we performed cross-validation 1000x (with participants randomly assigned to folds in order to avoid bias in fold assessment). These results are plotted in black for each model. To estimate a null distribution, we repeated this procedure with randomly shuffled ratings (plotted in gray). Boxplots show median, 25th and 75th percentiles (hinges) and 1.5*IQR (whiskers); dots indicate outliers.

6. Is there a figure legend for the supplemental figure? It is quite challenging to interpret the figure and to know which brain regions are being represented in the scatterplots.

We apologize for the confusion. We have revised this figure (now Supplementary Figure 3) to increase the size of the text and expanded the figure legend to spell out abbreviated brain regions.

7. Figure 3 (specificity analyses) might be better in the supplement since these are really just null, confirmatory results. Also, it is a bit confusing to have the figure 3 legend refer to results from Figure 2.

We appreciate the reviewer's comment and have revised the figure (now Fig. 4; reproduced above as Fig. R4). We consider these specificity analyses very important for the conclusions of the paper. Following comments from Reviewer #2, we conducted an additional analysis that revealed a novel double dissociation, whereby hippocampal networks predicted feelings of stress (not focus) and our control network predicted focus (not feelings of stress). Accordingly, we have pared down the figure presented in the main text (see Fig. R4) and present results from all predictive models in the supplement (new Supplementary Fig. 2).

8. In figure 4, it is challenging to understand the inset dots in the context of the figure. Could this information be included in a table instead? It would also be helpful to know the number of models predicting the cluster out of how many models (ie. the percentage).

We agree with this suggestion and have modified this figure (now Fig. 5; Fig. R10 below) accordingly, and added information about the percent of models to the text (p. 10). We also created a supplementary table that lists the MNI coordinates of all overlapping clusters across seed maps (new Supplementary Table 2). We believe it is important to represent these clusters as a figure, as it provides important anatomical information about the identified clusters that isn't provided in the whole brain cartoon plots shown in the other main text figures (e.g., Fig. 2b,e,h).

Fig. R10. Consistency of predictive hippocampal connectivity networks. **a** Clusters identified on every LOO fold as predictive across a significant number of models. Clusters shown separately for models predicting higher (positive network) and lower (negative network) stress responses. Inset dots indicate the number of models predicting arousal, stress, and PSS that included that cluster.

b Overlapping clusters predicting the same stress construct across hippocampal seeds.

References

- Fair DA, Miranda-Dominguez O, Snyder AZ et al. Correction of respiratory artifacts in MRI head motion estimates. *Neuroimage* **208**, 116400 (2020).
- Goldfarb EV, Seo D, Sinha R. Sex differences in neural stress responses and correlation with subjective stress and stress regulation. *Neurobiol Stress* **11**, 1100177 (2019).
- Gonzalez-Castillo J, Hoy CW, Handwerker DA, Robinson ME, Buchanan LC, Saad ZS, Bandettini PA. Tracking ongoing cognition in individuals using brief, whole-brain functional connectivity patterns. *PNAS* **28**, 8762-8767 (2015).
- Hunt LT, et al. Triple dissociation of attention and decision computations across prefrontal cortex. *Nat Neurosci* **21**, 1471-1481 (2018).
- Mickels JA, Fredrickson BL, Larkin GR, Lindberg CM, Maglio S, Reuter-Lorenz PA. Emotional category data on images from the international affective picture system. *Behav Res Meth* **37**, 626-630 (2005).
- Rosenberg MD, Finn ES, Constable RT, Chun MM. Predicting moment-to-moment attentional state. *Neuroimage* **114**, 249-256 (2015).
- Rosenberg MD, et al. Functional connectivity predicts changes in attention observed across minutes, days, and months. *PNAS* (2020).

Rudolph MD, et al. Maternal IL-6 during pregnancy can be estimated from newborn brain connectivity and predicts future working memory in offspring. *Nat Neurosci* 21, 765-772 (2018).

Sinha R, Lacadie CM, Constable RT, Seo D. Dynamic neural activity during stress signals resilient coping. *Proc Natl Acad Sci U S A* 113, 8837-8842 (2016).

Tompary A, Duncan K, Davachi L. Consolidation of Associative and Item Memory Is Related to Post-Encoding Functional Connectivity between the Ventral Tegmental Area and Different Medial Temporal Lobe Subregions during an Unrelated Task. *J Neurosci* **35**, 7326-31. (2015).

Tompary A, Al-Aidroos N, Turk-Browne NB. Attending to What and Where: Background Connectivity Integrates Categorical and Spatial Attention. *J Cogn Neurosci* **30**, 1281-1297. (2018)

Reviewers' Comments:

Reviewer #1:

Remarks to the Author:

The authors answered my questions satisfactorily.

Reviewer #2:

Remarks to the Author:

Goldfarb and colleagues have now done a number of additional analyses and provided a much richer description of their methods. Overall the revised manuscript is much improved, and I believe that it is an impressive contribution.

That said, a couple of interpretations that the authors have made remain problematic in my mind, and I believe that these issues should be addressed.

While the description of the task was much more clear in this version, it would be helpful if the authors could provide a better description of the different epochs within the task and exactly how those epochs are used for the presented analyses.

The authors make inferences about causation (eg. "attenuated stress") based on observed correlations. It would be useful if the authors could include a paragraph in the discussion describing alternative non-causal explanations.

Connectivity terminology suggests that individuals who differed in their hippocampal connectivity had different levels of stress attenuation. In my previous review I noted that this could also be driven by individual differences in univariate effects – for example, some subjects having hippocampus more active to stressful stimuli, and others have it less reactive to stressful stimuli. This would lead to differential brain correlations across these individuals without having any consistent hippocampal response across subjects (eg. lack of inter-subject correlations). But the interpretation of this mechanism would be much different from the one that the authors prefer – in which it is connectivity driving the effects, not "which" stimuli tend to drive hippocampal responses within an individual.

Similarly, the attention rebuttal looked for consistent relationships between the self reported attention and stressor/neutral condition across participants – but the potential for attentional confound, as I understand it, would be at the individual level... with some subjects paying more attention to the stressor condition, and others paying less.

Overall I find this work to be really compelling, my remaining issues are things that could be addressed by changing text. In particular it would be useful if the authors could clarify the last task detail requested and provide a more balanced discussion of alternative interpretations of their individual differences in connectivity findings.

Reviewer #3:

Remarks to the Author:

The authors have provided very thorough and thoughtful responses to the previous concerns. The additional analyses and revisions to the text have been helpful in addressing the concerns.

Response to Reviewer #2:

Goldfarb and colleagues have now done a number of additional analyses and provided a much richer description of their methods. Overall the revised manuscript is much improved, and I believe that it is an impressive contribution.

That said, a couple of interpretations that the authors have made remain problematic in my mind, and I believe that these issues should be addressed.

We appreciate the reviewer's careful reading and positive feedback on the revisions. All changes to the manuscript are indicated using the tracked changes feature.

1. While the description of the task was much more clear in this version, it would be helpful if the authors could provide a better description of the different epochs within the task and exactly how those epochs are used for the presented analyses.

The epochs are illustrated in Figure 1a, and we have revised the legend to Figure 1 to further clarify epochs during the task (p. 3).

We also describe the use of epochs in analyses in detail in the Methods (p. 15; see "fMRI preprocessing", "Seed-based connectivity maps", and "sCPM inputs: Feelings of stress").

2. The authors make inferences about causation (eg. "attenuated stress") based on observed correlations. It would be useful if the authors could include a paragraph in the discussion describing alternative non-causal explanations.

We agree that caution should be used when proposing causal interpretations. We have revised the language in the legend to Figure 3, the results (p. 7) and discussion (p. 13) accordingly to emphasize that the results are *consistent with* causal interpretations like attenuation.

3. Connectivity terminology suggests that individuals who differed in their hippocampal connectivity had different levels of stress attenuation. In my previous review I noted that this could also be driven by individual differences in univariate effects – for example, some subjects having hippocampus more active to stressful stimuli, and others have it less reactive to stressful stimuli. This would lead to differential brain correlations across these individuals without having any consistent hippocampal response across subjects (eg. lack of inter-subject correlations). But the interpretation of this mechanism would be much different from the one that the authors prefer – in which it is connectivity driving the effects, not "which" stimuli tend to drive hippocampal responses within an individual.

We agree that this is possible and have expanded our discussion of this point (p. 13). In particular, we consider that the relationship between hippocampal connectivity and subjective stress could be due to a separate factor modulating hippocampal connectivity (such as univariate responses) or a factor that

modulates both brain and subjective responses (such as attention; see #4 below). We also note that the use of background connectivity methods to assess functional hippocampal connectivity should mitigate the influence of stimulus-induced univariate responses, and that our findings from subjective attention ratings indicate that these are both uncorrelated with subjective stress and associated with a non-hippocampal network. Nevertheless, we emphasize that it is important to consider non-causal interpretations on p. 12

4. Similarly, the attention rebuttal looked for consistent relationships between the self reported attention and stressor/neutral condition across participants – but the potential for attentional confound, as I understand it, would be at the individual level... with some subjects paying more attention to the stressor condition, and others paying less.

We appreciate the reviewer's concerns about attention. In addition to our response to #3 above, we would like to emphasize that, on average, participants reported being highly focused on all images, and this did not differ between conditions (Stressor vs. Neutral: $t_{59} = 1.54$, $p = .13$). As the reviewer suggests, however, it is possible that some individuals paid more attention to the stressor condition than others. Critically, even if this were the case and some participants paid more attention to the stressor condition than others, this would not undermine our primary result that hippocampal functional connectivity predicts subjective stress in novel individuals. Rather, it could help explain *why* some participants felt more stressed than others. Likewise, other factors, such as individual differences in participants' emotional reactivity or rumination during the task, could help explain individual differences in subjective stress. We are excited to characterize factors that contribute to individual differences in hippocampal functional connectivity and subjective stress in future work.

Overall I find this work to be really compelling, my remaining issues are things that could be addressed by changing text. In particular it would be useful if the authors could clarify the last task detail requested and provide a more balanced discussion of alternative interpretations of their individual differences in connectivity findings.

We appreciate the reviewer's thoughtful comments and have addressed these points through clarification of task design and analyses as well as careful consideration in the discussion.